Secondary organic aerosol enhanced by increasing atmospheric oxidizing capacity in Beijing-Tianjin-Hebei (BTH), China

Tian Feng[1,2,6], Shuyu Zhao[1,3], Naifang Bei[4], Jiarui Wu[1,3], Suixin Liu[1,3], Xia Li[1,3], Lang Liu[1,3], Yang Qian[7], Qingchuan Yang[3], Yichen Wang[3], Weijian Zhou[1,5,6], Junji Cao[1,3,5], Guohui Li[1,3,5*]

[1]State Key Laboratory of Loess and Quaternary Geology, Institute of Earth Environment, Chinese Academy of Sciences, Xi'an, China
[2]Department of Geography & Spatial Information Techniques, Ningbo University, Ningbo, China
[3]Key Laboratory of Aerosol Chemistry and Physics, Institute of Earth Environment, Chinese Academy of Sciences, Xi'an, China
[4]School of Human Settlements and Civil Engineering, Xi'an Jiaotong University, Xi'an, China
[5]CAS Center for Excellence in Quaternary Science and Global Change, Xi'an, China
[6]Xi'an Accelerator Mass Spectrometry Center, Xi'an, China
[7]State Key Laboratory of Environmental Criteria and Risk Assessment & Environmental Standards Institute, Chinese Research Academy of Environmental Sciences, Beijing, China

*Correspondence to*: Guohui Li (ligh@ieecas.cn)

**Abstract**. The implementation of the Air Pollution Prevention and Control Action Plan in China since 2013 has profoundly altered the ambient pollutants in the Beijing-Tianjin-Hebei region (BTH). Here we show observations of substantially increased $O_3$ concentrations (about 30%) and a remarkable increase in the ratio of organic carbon (OC) to elemental carbon (EC) in BTH during the autumn from 2013 to 2015, revealing an enhancement in atmospheric oxidizing capacity (AOC) and secondary organic aerosol (SOA) formation. To explore the impacts of increasing AOC on the SOA formation, a severe air pollution episode from 3 to 8 October 2015 with high $O_3$ and $PM_{2.5}$ concentrations is simulated using the WRF-Chem model. The model performs reasonably well in simulating the spatial distributions of $PM_{2.5}$ and $O_3$ concentrations over BTH and the temporal variations of $PM_{2.5}$, $O_3$, $NO_2$, OC, and EC concentrations in Beijing compared to measurements. Sensitivity studies show that the change in AOC substantially influences the SOA formation in BTH. A sensitivity case characterized by a 31% $O_3$ decrease (or 36% OH decrease) reduces the SOA level by about 30% and the SOA fraction in total organic aerosol by 17% (from 0.52 to 0.43, dimensionless). Spatially, the SOA decrease caused by reduced AOC is ubiquitous in BTH, but the spatial relationship between SOA concentrations and the AOC is dependent on the SOA precursor distribution. Studies on SOA formation pathways further show that, when the AOC is reduced, the SOA from oxidation and partitioning of semi-volatile POA and co-emitted intermediate volatile organic compounds (IVOCs) decreases remarkably, followed by those from anthropogenic and biogenic VOCs. Meanwhile, the SOA decrease in the irreversible uptake of glyoxal and methylglyoxal on aerosol surfaces is negligible.

## 1 Introduction

Severe haze pollutions characterized by exceedingly high concentrations of fine particulate matter ($PM_{2.5}$) in Beijing-Tianjin-Hebei (BTH), China have drawn much attention from the public, government, and science community (Han et al., 2014; Jiang et al., 2015; Li et al., 2017a; Quan et al., 2014; Wang et al., 2016a). Elevated levels of $PM_{2.5}$ concentrations not only deteriorate air quality and visibility (Cao et al., 2012a; Feng et al., 2016; Seinfeld and Pandis, 2006), but also threat the public health and ecosystem (Cao et al., 2012b; Tie et al., 2016). In addition, $PM_{2.5}$ also modulates the energy budget of the earth system directly through absorbing and scattering the incident solar radiation and indirectly via acting as cloud condensation nuclei (CCN) and ice nuclei (IN) and subsequently altering cloud albedo and lifetime (Li et al., 2008; 2009; Wang et al., 2013; 2016b; 2018; Zhou et al., 2017).

Organic aerosol (OA) is a key component of $PM_{2.5}$ in ambient air, constituting of 20~90% of the $PM_{2.5}$ mass concentration (Kanakidou et al., 2005; Zhang et al., 2007). Previous studies have confirmed a large mass fraction of OA in ambient $PM_{2.5}$ in various Chinese cities. For example, Huang et al. (2014) have reported that OA accounts for 30~50% of the total $PM_{2.5}$ mass in megacities in China (e.g., Beijing, Shanghai, Guangzhou, and Xi'an) during severe haze episodes. Positive matrix factorization (PMF) analyses of the aerosol chemical speciation monitor (ACSM) measurements in Beijing have shown that OA contributes 40% and 52% of refractory submicron particulate matters in summer 2011 and winter 2012, respectively (Sun et al., 2013; 2012). Over BTH, Huang et al. (2017) have demonstrated that OA constitutes the most important part in the major chemical components of gravimetric $PM_{2.5}$ (20~25%) based on measurements at 3 urban sites from June 2014 to April 2015.

OA is traditionally categorized into primary and secondary organic aerosols (referred to as POA and SOA, respectively) in terms of its source and formation in the atmosphere. POA is the OA directly emitted into the atmosphere, and SOA is formed through a series of

chemical conversions of precursors and gas-particle partitioning, closely associated with the abundance of oxidants in the atmosphere and ambient temperature (Feng et al., 2016; Li et al., 2011b; Tsimpidi et al., 2010). SOA precursors mainly include volatile organic compounds (VOCs) emitted from anthropogenic and biogenic sources (Odum et al., 1996; Pankow, 1994), primary organic gases (POG) emitted or formed in the evaporation of POA, and intermediate VOCs (IVOCs) co-emitted with the POA (Lipsky and Robinson, 2006; Robinson et al., 2007; Shrivastava et al., 2006). The pathway of SOA formation is illustrated as follows (Li et al., 2011b; Robinson et al., 2007):

$$VOCs_{(g)} + oxidants \rightarrow OVOCs_{(g)} \leftrightarrow SOA_{(p)} \tag{R1}$$

$$IVOCs_{(g)} + OH \rightarrow OIVOCs \leftrightarrow SOA_{(p)} \tag{R2}$$

$$POA_{(p)} \leftrightarrow POG_{(g)} + OH \rightarrow OPOG_{(g)} \leftrightarrow SOA_{(p)} \tag{R3}$$

where the subscript $g$ and $p$ denote gas- and particle-phase, respectively. OVOCs, OIVOCs, and OPOG are oxidized VOCs, IVOCs, and POG, respectively. The oxidants in the chemical reactions mainly include ozone ($O_3$), hydroxyl radical (OH) and nitrate radical ($NO_3$). Apparently, the abundance of oxidants in the atmosphere plays an important role in the SOA formation, and increasing oxidants potentially enhance SOA formation. It is worth noting that heterogeneous reactions also play a considerable role in SOA formation (Fu et al., 2009; Li et al., 2011b).

Over the last decade, $O_3$ concentrations have dramatically increased in eastern China. For example, Cheng et al. (2016) have reported an increasing trend of the daily maximum 1h $O_3$ concentration over Beijing from 2004 to 2015. Ma et al. (2016) have observed a significant increase of surface $O_3$ concentrations at a rural station in the north of eastern China from 2003 to 2015. Since 2013, the implementation of the Air Pollution Prevention and Control Action Plan (APPCAP) in China have profoundly altered the air pollutants in BTH (He et al., 2017; Li et al., 2017b; Wu et al., 2017). He et al. (2017) have reported that the ambient OA

concentration has been significantly reduced by 27.5%, 17.4%, and 14.0% in Beijing, Tianjin,
and Hebei, respectively, from 2013 to 2017. The increasing $O_3$ concentration has become a
new culprit for the deterioration of the air quality in eastern China (Li et al., 2017b; Wu et al.,
2017). Li et al. (2017b) have reported that the $O_3$ concentration has increased by 10% from
2013 to 2015 averaged over 65 cities of eastern China during April ~ September. In Beijing,
the summertime $O_3$ concentration has increased by 23% from 2013 to 2015 (Wu et al., 2017).
Such an increasing trend of $O_3$ concentrations reflects an enhancement of the atmospheric
oxidizing capacity (AOC), which, as expected, potentially influences the SOA formation and
OA components. Therefore, it is imperative to evaluate the impact of the increasing AOC on
the SOA formation over BTH.
The objective of this study is to evaluate the impact of the increasing AOC on OA
components in BTH (Figure 1) during a haze episode in the autumn of 2015 using the
WRF-Chem model. Model and configuration are described in Sect. 2; the results and
discussion are presented in Sect. 3. The conclusions are drawn in Sect. 4.

**2   Model and method**
**2.1  WRF-Chem model and configuration**
In this study, simulations are performed using a specific version of the WRF-Chem
model (Grell et al., 2005) developed by Li et al. (2011b; 2011a; 2012; 2010). The model
includes a flexible gas phase chemical module and the Models-3 community multiscale air
quality (CMAQ) aerosol module (Binkowski and Roselle, 2003). The photolysis rates are
calculated using the FTUV module (Li et al., 2005; Tie et al., 2003) which takes into account
the effects of clouds and aerosols on photochemistry. A non-traditional SOA module based on
the volatility basis-set (VBS) method (Donahue et al., 2006; Robinson et al., 2007) is
incorporated into the model to simulate organic aerosols. In this module, POA is distributed

in logarithmically spaced volatility bins and presumed to be semi-volatile and photochemically reactive (Li et al., 2011b). The module uses 9 surrogate species with saturation concentration ranging from $10^{-2}$ to $10^6$ µg m$^{-3}$ at room temperature to represent POA compositions (Shrivastava et al., 2008). IVOCs, co-emitted with the POA but in the gas phase, are also oxidized by OH to form SOA. In addition, the SOA formation from glyoxal and methylglyoxal is included in the module, which is parameterized as a first-order irreversible uptake on aerosol surface with a reactive uptake coefficient of $3.7 \times 10^{-3}$ (Volkamer et al., 2007; Zhao et al., 2006). Inorganic aerosols are calculated by the ISORROPIA version 1.7 (Nenes et al., 1998). The GOCART (Georgia Tech/Goddard Global Ozone Chemistry Aerosol Radiation and Transport model) dust module is used to estimate the emission, transport, dry deposition, and gravitational settling of dust (Ginoux et al., 2001). The biomass burning emissions are from the Fire Inventory from NCAR (FINN) (Wiedinmyer et al., 2011; 2006). The dry deposition of chemical species is parameterized following Wesely (1989) and the wet deposition is calculated using the method in the CMAQ module (Binkowski and Roselle, 2003). Specifically, the surface and upper air observational wind fields from China Meteorological Administration (CMA) during the study period are assimilated using the four-dimensional data assimilation (FDDA) method to better simulate meteorological fields.

A severe haze episode from 3 to 8 October 2015 in BTH with high $O_3$ and $PM_{2.5}$ concentrations is simulated. The model is configured with one single domain which is centered at 116°E and 38°N with grid spacing of 6 km×6 km (200×200 grid cells). Thirty-five stretched vertical levels with spacing ranging from about 30 m near surface, to 500 m at 2.5 km, and 1 km above 14 km are used in the model configuration. The monthly average anthropogenic emission inventory is developed by Zhang et al. (2009) and Li et al. (2017c), with the base year of 2013, including agriculture, industry, power generation,

residential, and transportation sources. The temporal resolution of emissions used in
simulations is 1 hour, and the temporal allocation for different sources follows those in Zhang
et al. (2009). Figure 2 presents the spatial distributions of anthropogenic volatile organic
compounds (VOCs) and organic carbon (OC) emissions in October, showing high emissions
in urban areas. The emissions of various species in Beijing, Tianjin, Hebei, and the entire
domain in October 2015 are summarized in Table 1. Biogenic emissions are calculated online
using the MEGAN (Model of Emissions of Gases and Aerosol from Nature) model (Guenther
et al., 2006). The model configuration is presented in Table 2.
**2.2  Pollutant measurements**
Measurement data used in this study include the hourly concentrations of $O_3$, $NO_2$, $SO_2$,
CO, and $PM_{2.5}$ from ambient monitoring stations of China's Ministry of Environment and
Ecology (China MEE) and hourly OC and EC concentrations in $PM_{2.5}$ measured hourly at
Chinese Research Academy of Environmental Sciences (CRAES) using a Sunset OC/EC
Analyzer (RT-4, Sunset Lab, USA). The OC/EC analyzer has been widely used in ambient
and indoor OC/EC detections in China (Liu et al., 2018; Wei et al., 2014). In addition, hourly
submicron POA and SOA concentrations are obtained from the ACSM measurement
analyzed using the PMF method at National Center for Nanoscience and Technology (NCNT)
in Beijing.
**2.3  Model simulations**
We define the simulation with the AOC in October 2015 as the reference (REF). The
model result in REF is compared with the observations to evaluate the model performance.
To examine the impact of increasing AOC on OA components, we perform 4 sensitivity
experiments (SEN1~4) by varying AOC. Compared with the REF simulation, we decrease all
the photolysis frequencies by 10%, 20%, 30%, and 40%, respectively, in the model
simulations.
**2.4 Statistic method**
The mean bias (MB), normalized mean bias (NMB), root mean square error (RMSE),
index of agreement (IOA), and linear Pearson correlation coefficient (*r*) are selected to
evaluate the WRF-Chem model simulations against observations.
$MB = \frac{1}{N}\sum_{i=1}^{N}(P_i - O_i)$         (1)
$NMB = \frac{\sum_{i=1}^{N}(P_i - O_i)}{\sum_{i=1}^{N}O_i} \times 100\%$     (2)
$RMSE = \left[\frac{1}{N}\sum_{i=1}^{N}(P_i - O_i)^2\right]^{\frac{1}{2}}$     (3)
$IOA = 1 - \frac{\sum_{i=1}^{N}(P_i - O_i)^2}{\sum_{i=1}^{N}(|P_i - \bar{O}| + |O_i - \bar{O}|)^2}$     (4)
$r = \frac{\sum_{i=1}^{N}(P_i - \bar{P})(O_i - \bar{O})}{\sqrt{\sum_{i=1}^{N}(P_i - \bar{P})^2}\sqrt{\sum_{i=1}^{N}(O_i - \bar{O})^2}}$     (5)
where $P_i$ and $O_i$ are the simulated and observed variables, respectively. $N$ is the total
number of predictions. $\bar{P}$ and $\bar{O}$ denote the average of predictions and observations,
respectively. IOA ranges from 0 to 1 theoretically, with 1 suggesting perfect agreement
between predictions and observations.

181**2** **3**    **Results and discussion**

**3.1 Observed increasing O₃ concentration and OC/EC ratio**
Figure 3a shows the annual variation of measured mean concentrations of $O_3$, $NO_2$, $SO_2$,
CO, and $PM_{2.5}$ over BTH in the autumn from 2013 to 2017. To better present the pollution
characteristics in autumn, the observations from 15 September to 14 November are selected
in this study, which avoids the heating period (starting from 15 November) in northern China.
Obvious decreasing trends in $NO_2$, $SO_2$, CO, and $PM_{2.5}$ concentrations during recent years
are observed since the implementation of APPCAP (Figure 3a). The $O_3$ concentration,
however, has increased by about 30% during the same period. Such an $O_3$ increase indicates a
considerable enhancement of the AOC, considering the controlling role of $O_3$ in the AOC in
the troposphere (Brasseur et al., 1999). The reason for the AOC or $O_3$ increase since 2013
still remains elusive. Li et al. (2018) have proposed that the $O_3$ increase in China since 2013
is associated with the decreased removal efficiency of $HO_x$ (OH + peroxy) on aerosol
surfaces caused by the reduced aerosol concentrations since the implementation of APPCAP.
However, further studies need to be conducted to evaluate the $O_3$ contribution of the
photolysis change caused by the aerosol-radiation interaction and aerosol-cloud interaction
induced by decreasing aerosols in China.

As important $PM_{2.5}$ components, organic carbon (OC) and elemental carbon (EC) are

measured in Beijing in the autumns from 2013 to 2015 (Figure 3b). The measured OC/EC
ratio has substantially increased during the 3 years (about 44%). There are two possible
reasons for this. Firstly, the increase might be attributed, at least in part, to the increasing
AOC, which enhances the SOA formation. Secondly, the increase might be caused by the
changes in OC and EC emissions due to the implementation of APPCAP. The large
variability in OC and EC concentrations in Figure 3b suggests considerable emission changes,
although it is still difficult to evaluate exactly how much the emissions have been changed.
Since SOA formation is closely associated with the abundance of oxidants (Li et al., 2011b;
Robinson et al., 2007), OA in BTH is expected to be more oxygenated (add oxygen) and
hence to increase in mass with enhanced AOC.
**3.2  Model evaluation**
**3.2.1 Meteorological fields**

Model performance in meteorological fields is crucial for the simulation of air pollutants

(Bei et al., 2010; 2012; 2017). Figures 4 and 5 present the simulated temporal variations of
the surface temperature, relative humidity, wind speed, and wind direction against
observations at 4 meteorological stations (Beijing, Tianjin, Shijiazhuang, and Baoding,
Figure 1) in BTH.
The model performs well in reproducing the observed diurnal cycles of the surface
temperature and relative humidity at the 4 stations (Figure 4). The simulated temporal
variations of temperature and relative humidity are in good agreement with the observations
at the meteorological station in Tianjin. However, in the other 3 cities, the model generally
cannot well reproduce the fairly low temperature and high relative humidity during nighttime.
In general, the model replicates the low winds during 3-7 October in Beijing, Tianjin, and
Baoding, but slightly overestimates the wind speed in Shijiazhuang, particularly in the
morning of 8 October (Figure 5). The model fails to produce the occurrence of the strong
wind with a maximum speed exceeding 10 m s$^{-1}$ and the wind direction at noon on 8 October
in Beijing. Overall, the model still generally captures the temporal variations of wind
directions in the 4 cities.
**3.2.2 PM$_{2.5}$, O$_3$, and NO$_2$**
The spatial distributions of simulated and observed daily PM$_{2.5}$ concentration during the
period from 3 to 8 October 2015 are presented in Figure 6 along with wind fields. This haze
event in BTH can be divided into 4 stages: (1) startup (3 October), (2) development (4
October), (3) maturation (5-7 October), and (4) dissipation (8 October). On 3 October, the
haze formed in Shandong and southern BTH, accompanied by weak winds near the surface
(< 2 m s$^{-1}$). On 4 October, the southerly wind prevailed, causing trans-boundary transport of
air pollutants from south to north, and the PM$_{2.5}$ concentration in BTH rapidly increased.
From 5 to 7 October, the southerly wind continued and the haze became persistently severe in
BTH. Finally, a strong northerly wind cleaned up the haze within several hours on 8 October.
The model reasonably reproduces the haze evolution in BTH, but it generally underestimates
the PM$_{2.5}$ concentration in Shandong province.
Figure 7 shows the observed and simulated spatial distribution of peak O$_3$ concentrations

at 14:00 (local time, hereafter) during the episode. Similar to $PM_{2.5}$, the high $O_3$ concentration first occurred in Shandong province on 3 October, and propagated to BTH on 4 October. During the maturation period, the $O_3$ level in BTH still remained high. The simulated spatial distributions of $O_3$ concentrations generally agree well with the observations during the four stages, but underestimation or overestimation still exists.

Figure 8 shows the temporal variations of the simulated and observed surface $PM_{2.5}$, $O_3$, and $NO_2$ concentrations averaged over 12 ambient monitoring stations in Beijing. The simulated and observed $PM_{2.5}$ temporal pattern clearly shows the 4 stages of the haze episode, with the $PM_{2.5}$ concentration increasing from about 20 μg m$^{-3}$ during the startup stage to more than 300 μg m$^{-3}$ during the maturation stage. The model generally replicates the evolution of the observed $PM_{2.5}$ concentration with an IOA ($r$) of 0.95 (0.91), but slightly underestimates the $PM_{2.5}$ concentration with an MB (NMB) of -13.0 μg m$^{-3}$ (-8.7%). The simulated diurnal profile of the $O_3$ concentration is well consistent with observations, with an IOA ($r$) of 0.94 (0.92), but the model overestimates the $O_3$ diurnal lows during the maturation stage. Additionally, Figures 8a and 8b also show that both $O_3$ and $PM_{2.5}$ pollutions occur during the maturation stage in Beijing, as previously reported for non-winter seasons (Jia et al., 2017). The model also exhibits good performance in simulating the temporal variation of $NO_2$ concentrations, with an IOA ($r$) of 0.90 (0.81).

### 3.2.3 Carbonaceous aerosols

The simulated carbonaceous aerosols including POA, SOA, and EC in the model are compared with the hourly observations in Beijing (Figure 9). In general, the temporal variations of the measured carbonaceous aerosols are similar to that of the $PM_{2.5}$ in Figure 8a. The model yields the increasing trend of the POA concentration from the startup to maturation stages compared to the measurements, but cannot well capture the observed spiky peaks, with an IOA ($r$) of 0.75 (0.58). Figure 9b shows that the observed SOA concentration

is remarkably enhanced during the maturation stage, ranging from 30 to 90 μg m$^{-3}$, which is
well predicted by the model. The MB, NMB, IOA, and $r$ for the simulated SOA concentration
are -2.1 μg m$^{-3}$, -6.9%, 0.89, and 0.81, respectively. Although the IOA and $r$ for the simulated
EC concentration reach 0.92 and 0.90, respectively, the model considerably underestimates
the EC concentration against measurement on October 6 and 7, which is likely caused by the
variation in the anthropogenic emissions.
Overall, the model performs reasonably in reproducing the observed meteorological
fields and the PM$_{2.5}$ and O$_3$ evolutions, and temporal variations of carbonaceous aerosols
during the pollution episode, providing a reliable base for the further sensitivity studies.
**3.3 Impact of increasing oxidizing capacity on OA**
Compared to the REF simulation, when the photolysis frequencies are decreased by 10%,
20%, 30%, and 40% in the 4 sensitivity experiments (SEN1~4), respectively, the O$_3$ (OH
radical) concentration is correspondingly reduced by 7.4% (9.2%), 15.1% (18.3%), 22.9%
(26.9%), and 30.9% (35.7%). It is worth noting that the REF experiment is assumed to
represent a situation in autumn with the high AOC, and the SEN1~4 experiments could be
regarded as 4 scenarios with the different lower AOC.
**3.3.1 OA component changes**
Figures 10a and 10b show the variation of POA, SOA, and TOA concentrations as a
function of the O$_3$ and OH concentration changes over BTH by differentiating REF and the 4
sensitivity experiments, respectively. The SOA level decreases almost linearly with
decreasing O$_3$ or OH concentrations, indicating that the AOC plays an important role in the
SOA formation over BTH. In the SEN4 experiment with the most reduction of the AOC, the
SOA concentration in BTH is reduced by 31.3% or 5.2 μg m$^{-3}$ on average during the episode.
The reduction in POA level with decreasing O$_3$ and OH concentrations is generally not
substantial, indicating that IVOCs and VOCs are the most important SOA contributors. The
TOA (sum of POA and SOA) level also exhibits a decreasing trend with $O_3$ and OH
concentrations. In Figures 10c and 10d, the SOA mass fraction in TOA and OC/EC ratio is
considerably reduced as the AOC decreases. The SOA fraction (OC/EC ratio) is about 0.52
(6.39) in the REF simulation and almost linearly decreases to around 0.43 (5.49) in the SEN4
experiment, indicating a slower aging process of OA with decreasing AOC. The simulated
decrease in OC/EC ratio due to reduced AOC could interpret the observed change in OC/EC
ratio in Figure 3b to some degree.
It is worth noting that the increase in OC/EC ratio potentially influences atmospheric
radiation and thermodynamical profiles, through enhancing aerosol scattering and absorption
simultaneously (Wang et al., 2013). When the photolysis frequencies are reduced by 30% in
the SEN3 experiment, compared to the REF, the downward shortwave radiation is reduced by
1.2 W $m^{-2}$ on average in BTH, and the surface temperature is decreased by around 0.016 $^oC$
during daytime. Effects of the AOC change on the temperature profile is not significant, and
the daytime temperature decrease in the SEN3 experiment is less than 0.005 $^oC$ within 1 km
height from surface.
**3.3.2 SOA spatial change in SEN3**
Among the 4 experiments, the $O_3$ change over BTH in SEN3 is close to the observed
change (about 30% increase, Figure 3a). Therefore, we further analyze the SOA spatial
change in the SEN3 experiment. To illustrate the impacts of the AOC change on the spatial
distribution of SOA concentrations in BTH during the haze episode, Figure 11 shows the
spatial distributions of the variation in the main oxidant (OH) and SOA concentrations
averaged from 4 to 7 October by differentiating REF and the SEN3 experiment. When the
photolysis rates are reduced by 30%, the OH concentration over BTH is generally decreased
by more than 20%, but the OH variation distribution is not uniform (Figure 11c). The OH
decrease is remarkable in the west of Hebei province, compared to the other regions of BTH,

showing the variety of OH sinks and its reservoirs. The most striking decrease in SOA mass occurs in Beijing and surrounding areas, exceeding 8 $\mu g$ $m^{-3}$ (Figure 11b); while the mass percentage decrease is more noticeable in the west of Hebei province (more than 26%, Figure 11d), which is generally corresponding to the OH reduction. Although OH is the main oxidant in the SOA formation during daytime, the spatial change of SOA concentration is not well consistent with that of the OH concentration, especially for the mass change (Figure 11a). The geographical difference probably results from the spatial distribution variation of anthropogenic and biogenic precursors of SOA. In the middle and east BTH, massive anthropogenic SOA precursors are emitted from residential, transportation and industrial sources; while in the west BTH, biogenic precursor emissions are dominant for the SOA formation, but much less than those from anthropogenic sources in the middle and east BTH (Figure 2).

**3.3.3 Changes in the secondary organic aerosol pathways**

The spatial decrease in SOA concentration highlights the important influence of AOC change on the SOA formation over BTH. To understand how the SOA components are affected by the changing AOC, we further examine the variation of the SOA formation pathways. The non-traditional SOA module employed in the WRF-Chem model includes 4 SOA formation pathways: oxidation and partitioning of (1) POA treated as semivolatile and co-emitted IVOCs (PSOA), (2) anthropogenic VOCs (ASOA), and (3) biogenic VOCs (BSOA), and (4) heterogeneous reactions of glyoxal and methylglyoxal on aerosol surfaces (HSOA) (Feng et al., 2016; Li et al., 2011b). SOA formation from the 4 pathways in the REF and 4 sensitivity experiments is analyzed to examine the influence of the changing AOC.

Figure 12 shows the changes in SOA concentrations in the 4 sensitivity experiments compared with the REF simulation in Beijing, Tianjin, and Hebei, respectively. The impact of AOC reduction on the 4 pathways and the resulting SOA decreases differ markedly from one

another. Since the oxidation and partitioning of semivolatile POA and co-emitted IVOCs
contribute the most to the SOA concentration (Feng et al., 2016), the most substantial SOA
decrease occurs in the PSOA, followed by the ASOA and BSOA. The decrease from the
HSOA in the 4 experiments is negligible, because the HSOA is mainly contributed by the
direct emission of glyoxal and methylglyoxal from residential combustion source (Xing et al.,

2019).

It should be noted that, the percentage decreases of SOA from the PSOA in Beijing,
Tianjin, and Hebei are comparable (Figure 12b, d, and f), although the mass decrease in
Beijing is apparently larger than that in Tianjin and Hebei, indicating the ubiquitous effect of
AOC on the PSOA over BTH. The SOA decrease from the ASOA in Beijing is more than that
in Tianjin and Hebei, which is likely due to higher and concentrated anthropogenic VOCs
emissions in Beijing. By contrast, the SOA decreases from the BSOA are all rather small, and
compared with Beijing and Hebei, the SOA decrease from BSOA in Tianjin is even less. In
the northwestern part of BTH, the widely distributed forests emit abundant VOCs; whereas
the biogenic VOC emissions in Tianjin are much lower owing to a less vegetation cover. In
the SEN3 experiment, the SOA decreases in the PSOA, ASOA, and BSOA are 28%, 8%, and
1% in Beijing, respectively, and slightly less than those in Tianjin and Hebei.

**3  4  Summary and conclusions**
Observations have revealed substantial increase in $O_3$ concentrations (about 30%) over
BTH and in the ratio of organic carbon (OC) to elemental carbon (EC) in Beijing during the
autumn from 2013 to 2015, indicating enhanced AOC and SOA formation. We simulate a
6-day haze episode in BTH from 3 to 8 October 2015 using the WRF-Chem model, as a case
study, to explore the influence of the increasing AOC on the SOA formation in BTH.
Generally, the model performs reasonably well in predicting the temporal variations of
the temperature, relative humidity, wind speed and direction at 4 meteorological stations in
BTH. The spatial distributions of $PM_{2.5}$ and $O_3$ concentrations over BTH and the temporal
variations of $PM_{2.5}$, $O_3$, $NO_2$, and carbonaceous aerosols including POA, SOA, and EC in
Beijing are also well reproduced against measurements.
Four sensitivity experiments with different reductions in the AOC show that changing
AOC substantially affects the SOA formation. In the SEN4 scenario, characterized by a 30.9%
(35.7%) decrease in $O_3$ (OH) concentration, the SOA concentration is reduced by 31.3% and
the SOA mass fraction in TOA is reduced from 0.52 to 0.43. Spatially, the SOA reduction is
ubiquitous over BTH, but the spatial relationship between the SOA concentration and AOC is
dependent on the SOA precursor distribution. Among the 4 pathways of the SOA formation
in the non-traditional SOA module, the largest SOA reduction in the reduced AOC
environment occurs in the PSOA, followed by the ASOA and BSOA. By contrast, the SOA
reduction in the HSOA is negligible.
Although the model reasonably reproduces the observed meteorological fields and
chemical species in BTH, model discrepancies still exist, especially for the $PM_{2.5}$ simulation
in Shandong. More studies need to be performed to improve the model simulation and
evaluate the impact of AOC change on SOA formation using more accurate meteorological
fields and updated anthropogenic emissions.


*Author contribution.* Guohui Li, as the contact author, provided the ideas and financial
support, verified the conclusions, and revised the paper. Tian Feng conducted a research,
designed the experiments, carried the methodology out, performed the simulation, processed
the data, prepared the data visualization, and prepared the manuscript with contributions from
all authors. Shuyu Zhao and Naifang Bei provided the treatment of meteorological data,

analyzed the study data, validated the model performance, and reviewed the manuscript. Suixin Liu, Yang Qian, Yichen Wang, and Qingchuan Yang provided the observation data used in the study, synthesized the observation, and reviewed the paper. Jiarui Wu, Xia Li, and Lang Liu analyzed the initial simulation data, visualized the model results and reviewed the paper. Weijian Zhou and Junji Cao provided critical reviews pre-publication stage.

*Acknowledgements*. This work is financially supported by the National Key R&D Plan (Quantitative Relationship and Regulation Principle between Regional Oxidation Capacity of Atmospheric and Air Quality (2017YFC0210000)) and National Research Program for Key Issues in Air Pollution Control (DQGG0105). Tian Feng is supported by National Natural Science Foundation of China (no. 41703127, 41430424, 41661144020).

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

Contributions of trans-boundary transport to summertime air quality in Beijing, China,
Atmos. Chem. Phys., 17(3), 2035–2051, doi:10.5194/acp-17-2035-2017, 2017.
Xing, L., Wu, J., Elser, M., Tong, S., Liu, S., Li, X., Liu, L., Cao, J., Zhou, J., Haddad, El, I.,
Huang, R., Ge, M., Tie, X., Prévôt, A. S. H. and Li, G.: Wintertime secondary organic
aerosol formation in Beijing–Tianjin–Hebei (BTH): contributions of HONO sources and
heterogeneous reactions, Atmos. Chem. Phys., 19(4), 2343–2359,
doi:10.5194/acp-19-2343-2019, 2019.
Zhang, Q., Jimenez, J. L., Canagaratna, M. R., Allan, J. D., Coe, H., Ulbrich, I., Alfarra, M.
R., Takami, A., Middlebrook, A. M., Sun, Y. L., Dzepina, K., Dunlea, E., Docherty, K.,
DeCarlo, P. F., Salcedo, D., Onasch, T., Jayne, J. T., Miyoshi, T., Shimono, A.,
Hatakeyama, S., Takegawa, N., Kondo, Y., Schneider, J., Drewnick, F., Borrmann, S.,
Weimer, S., Demerjian, K., Williams, P., Bower, K., Bahreini, R., Cottrell, L., Griffin, R.
J., Rautiainen, J., Sun, J. Y., Zhang, Y. M. and Worsnop, D. R.: Ubiquity and dominance
of oxygenated species in organic aerosols in anthropogenically-influenced Northern
Hemisphere midlatitudes, Geophys. Res. Lett., 34(13), L13801,
doi:10.1029/2007GL029979, 2007.
Zhang, Q., Streets, D. G. and Carmichael, G. R.: Asian emissions in 2006 for the NASA
INTEX-B mission, Atmos. Chem. Phys., 9(14), 5131–5153,
doi:10.5194/acp-9-5131-2009, 2009.
Zhao, J., Levitt, N. P., Zhang, R. and Chen, J.: Heterogeneous Reactions of Methylglyoxal in
Acidic Media:   Implications for Secondary Organic Aerosol Formation, Environ. Sci.
Technol., 40(24), 7682–7687, doi:10.1021/es060610k, 2006.
Zhou, X., Bei, N., Liu, H., Cao, J., Xing, L., Lei, W., Molina, L. T. and Li, G.: Aerosol
effects on the development of cumulus clouds over the Tibetan Plateau, Atmos. Chem.
Phys., 17(12), 7423–7434, doi:10.5194/acp-17-7423-2017, 2017.





Table 1 Anthropogenic emissions of various species in the simulation domain in October
2015 (Unit: Mton month$^{-1}$)

| Species | NOx | SO$_2$ | NH$_3$ | CO | VOC | OC | EC |
|---|---|---|---|---|---|---|---|
| Beijing | 0.31 | 0.02 | 0.05 | 0.66 | 1.51 | 0.03 | 0.01 |
| Tianjin | 0.24 | 0.09 | 0.05 | 0.09 | 2.8 | 0.05 | 0.01 |
| Hebei | 2.21 | 0.7 | 0.62 | 3.59 | 21.59 | 0.41 | 0.06 |
| Domain | 14.21 | 7.1 | 4.45 | 22.19 | 124.71 | 2.56 | 0.3 |


Table 2 WRF-Chem model configuration

| Item | Configuration |
|---|---|
| Period | 3 ~ 8 October 2015 |
| Regions | Beijing-Tianjin-Hebei, China |
| Domain center | 116°E, 38°N |
| Domain size | 1200 km × 1200 km |
| Horizontal resolution | 6 km × 6 km |
| Vertical resolution | 35 vertical levels with a stretched vertical grid with spacing ranging from 50 m near surface, to 500 m at 2.5 km and 1 km above 14 km |
| Microphysics scheme | WRF Single-Moment 6-class scheme (Hong and Lim, 2006) |
| Boundary layer scheme | MYJ TKE scheme (Janjić, 2002) |
| Surface layer scheme | MYJ surface scheme (Janjić, 2002) |
| Land-surface scheme | Noah land surface model (Chen and Dudhia, 2001) |
| Longwave radiation scheme | New Goddard scheme (Chou et al., 2001) |
| Shortwave radiation scheme | New Goddard scheme (Chou and Suarez, 1999) |
| Meteorological boundary and initial condition | NCEP 1° × 1° reanalysis data |
| Chemical boundary and initial condition | MOZART 6-h output (Horowitz et al., 2003) |
| Anthropogenic emission inventory | SAPRC99 chemical mechanism emissions (Zhang et al., 2009), base year: 2013 |
| Biogenic emission inventory | MEGAN model developed by Guenther et al. (2006) |
| Spin-up time | 1.5 days |


Table 3 Description of the reference simulation and sensitivity experiments

| Case ID | Description |
|---------|-------------|
| REF | The reference simulation constrained by observations |
| SEN1 | 10% decrease in photolysis frequencies |
| SEN2 | 20% decrease in photolysis frequencies |
| SEN3 | 30% decrease in photolysis frequencies |
| SEN4 | 40% decrease in photolysis frequencies |






Figure 1 Model domain with the topography. The black circles denote the locations of the cities with ambient air quality monitoring sites, and the size of the circles represents the number of sites in each city. The white triangles show the location of the meteorological stations in Beijing, Tianjin, Shijiazhuang, and Baoding. The light blue and pink dots in Beijing show the observation sites with the POA/SOA (NCNT) and OC/EC (CRAES) measurements, respectively.

Figure 2 Geographic distributions of anthropogenic emissions of (a) volatile organic compounds and (b) organic carbon in October in the simulation domain. The black lines present provincial boundaries in China.

Figure 3 (a) Measured concentrations of $O_3$, $NO_2$, $SO_2$, CO, and $PM_{2.5}$ in BTH averaged during 15 September ~ 14 November from 2013 to 2017, and (b) OC and EC concentrations (bars) and OC/EC ratios (line) measured in Beijing averaged during 15 September ~ 14 November from 2013 to 2015.

Figure 4 Simulated (red curves) and observed (black dots) temporal profiles of surface (a-d) temperature and (e-h) relative humidity in (a, e) Beijing, (b, f) Tianjin, (c, g) Shijiazhuang, and (d, h) Baoding from 3 to 8 October 2015.

Figure 5 Simulated (red curves) and observed (black dots) temporal profiles of surface (a-d) wind speed and (e-h) wind direction in (a, e) Beijing, (b, f) Tianjin, (c, g) Shijiazhuang, and (d, h) Baoding from 3 to 8 October 2015.

Figure 6 Spatial distributions of the modeled (colored shadings) and observed (colored dots) surface daily $PM_{2.5}$ concentration from 3 to 8 October 2015. Black arrows show the simulated surface winds.

Figure 7 Spatial distributions of the modeled (colored shadings) and observed (colored dots) surface $O_3$ concentration at 14:00 (local time) from 3 to 8 October 2015. Black arrows show the simulated surface winds.

Figure 8 Diurnal variations of the modeled (red curves) and observed (black dots) surface (a) $PM_{2.5}$, (b) $O_3$, and (c) $NO_2$ concentrations averaged over 12 ambient monitoring stations in Beijing from 3 to 8 October 2015.

Figure 9 Diurnal variations of the modeled (red curves) and observed (black dots) surface submicron (a) POA and (b) SOA concentrations at the NCNT station, and (c) EC concentration in $PM_{2.5}$ at the CRAES station in Beijing from 3 to 8 October 2015.

Figure 10 Impacts of changes in the AOC on organic aerosol components in BTH in 4 sensitivity experiments. (a) Concentration changes of POA, SOA, and TOA *versus* $O_3$, (b) Concentration changes of POA, SOA, and TOA *versus* OH, (c) SOA fraction in TOA *versus* OH concentration change, and (d) OC/EC ratio *versus* OH concentration change.

Figure 11 Spatial distributions of changes in (a, c) OH and (b, d) SOA concentrations averaged from 4 to 7 October 2015 in the SEN3 experiment compared to the REF simulation (SEN3 – REF).

Figure 12 Histogram showing the decreases of SOA from various pathways in (a, b) Beijing,
(c, d) Tianjin, and (e, f) Hebei in the sensitivity experiments compared to the REF
simulation (SENx – REF, x = 1, 2, 3, and 4). PSOA: oxidation and partitioning of
semivolatile POA and co-emitted IVOCs; ASOA: oxidation and partitioning of
anthropogenic VOCs; BSOA: oxidation and partitioning of biogenic VOCs; HSOA:
heterogeneous reactions of glyoxal and methylglyoxal on aerosol surfaces.


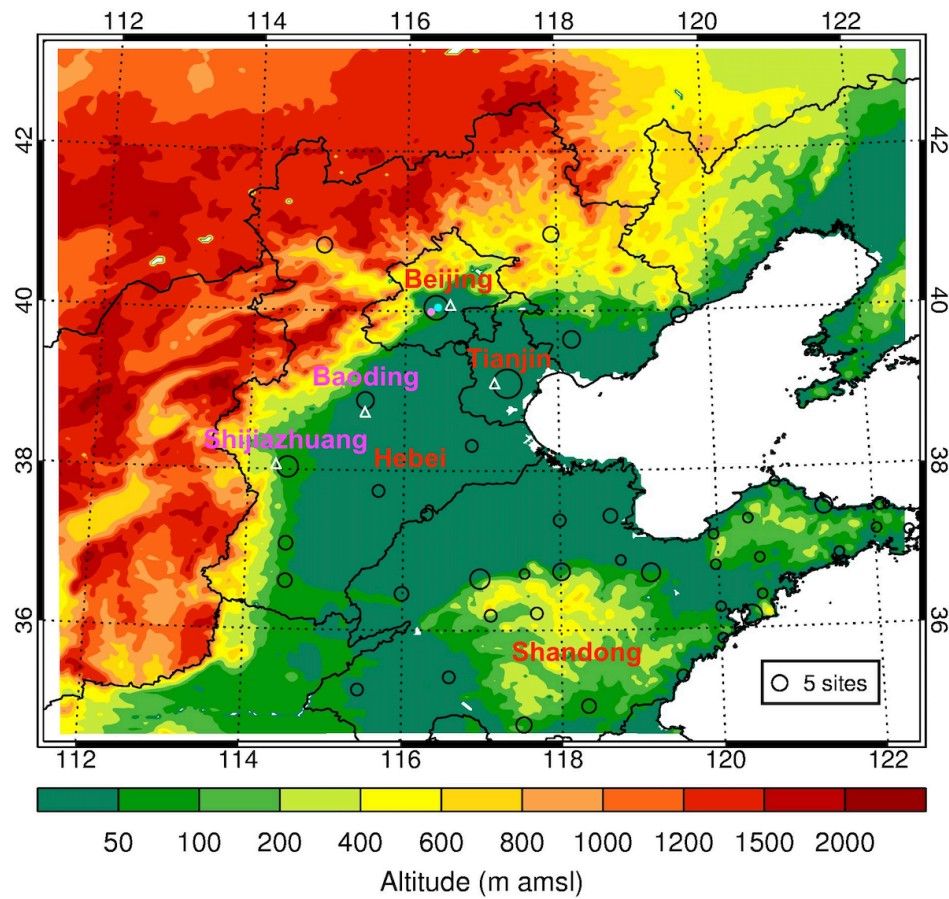

Figure 1 Model domain with the topography. The black circles denote the locations of the
cities with ambient air quality monitoring sites, and the size of the circles represents the
number of sites in each city. The white triangles show the location of the meteorological
stations in Beijing, Tianjin, Shijiazhuang, and Baoding. The light blue and pink dots in
Beijing show the observation sites with the POA/SOA (NCNT) and OC/EC (CRAES)
measurements, respectively.

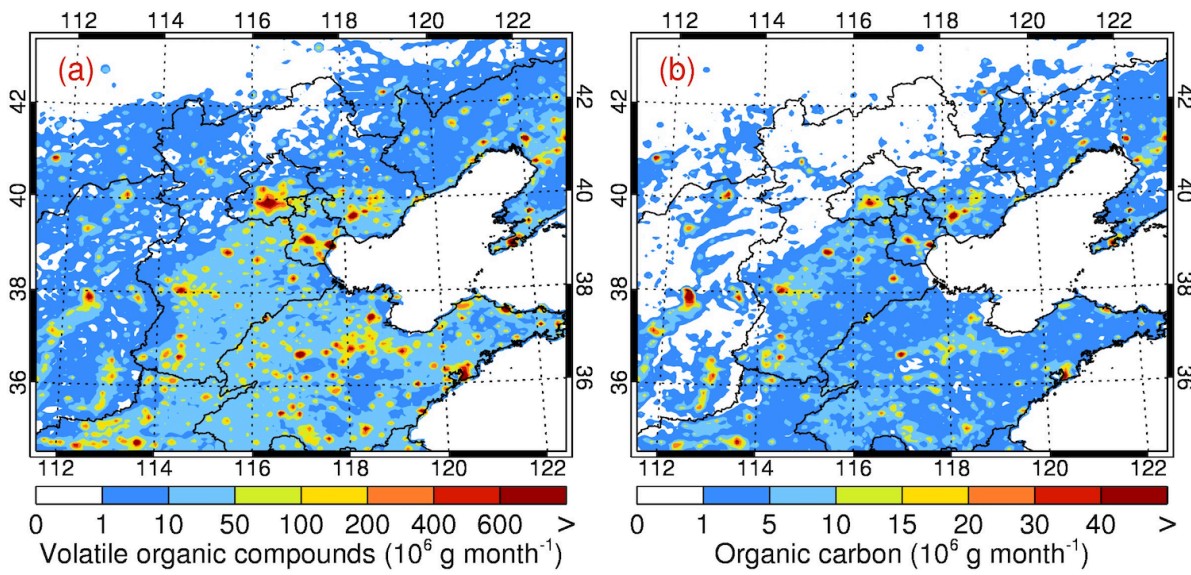



Figure 2 Geographic distributions of anthropogenic emissions of (a) volatile organic
compounds and (b) organic carbon in October in the simulation domain. The black lines
present provincial boundaries in China.

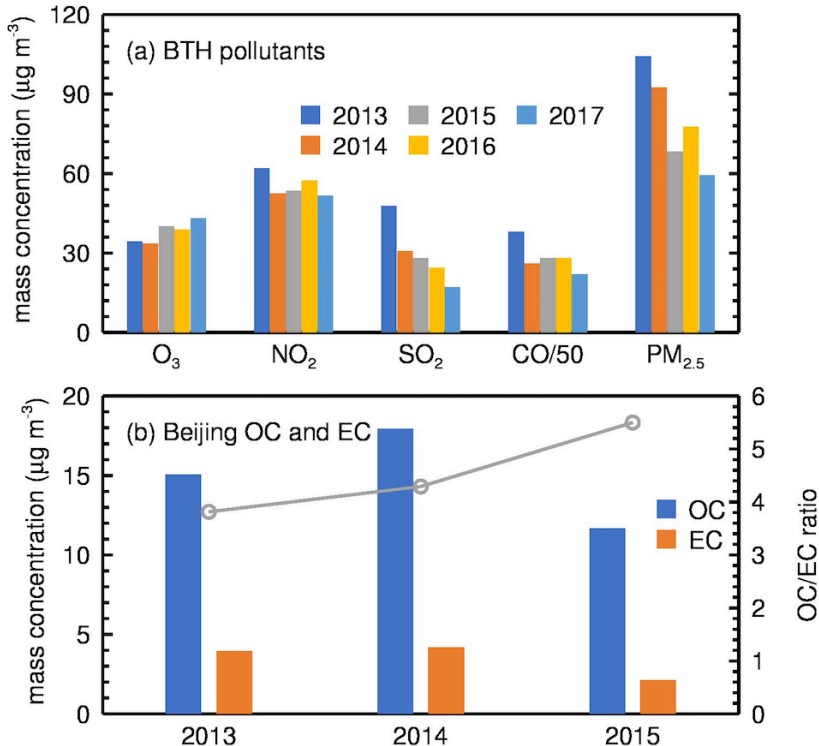

Figure 3 (a) Measured concentrations of $O_3$, $NO_2$, $SO_2$, CO, and $PM_{2.5}$ in BTH averaged
during 15 September ~ 14 November from 2013 to 2017, and (b) OC and EC concentrations
(bars) and OC/EC ratios (line) measured in Beijing averaged during 15 September ~ 14
November from 2013 to 2015.

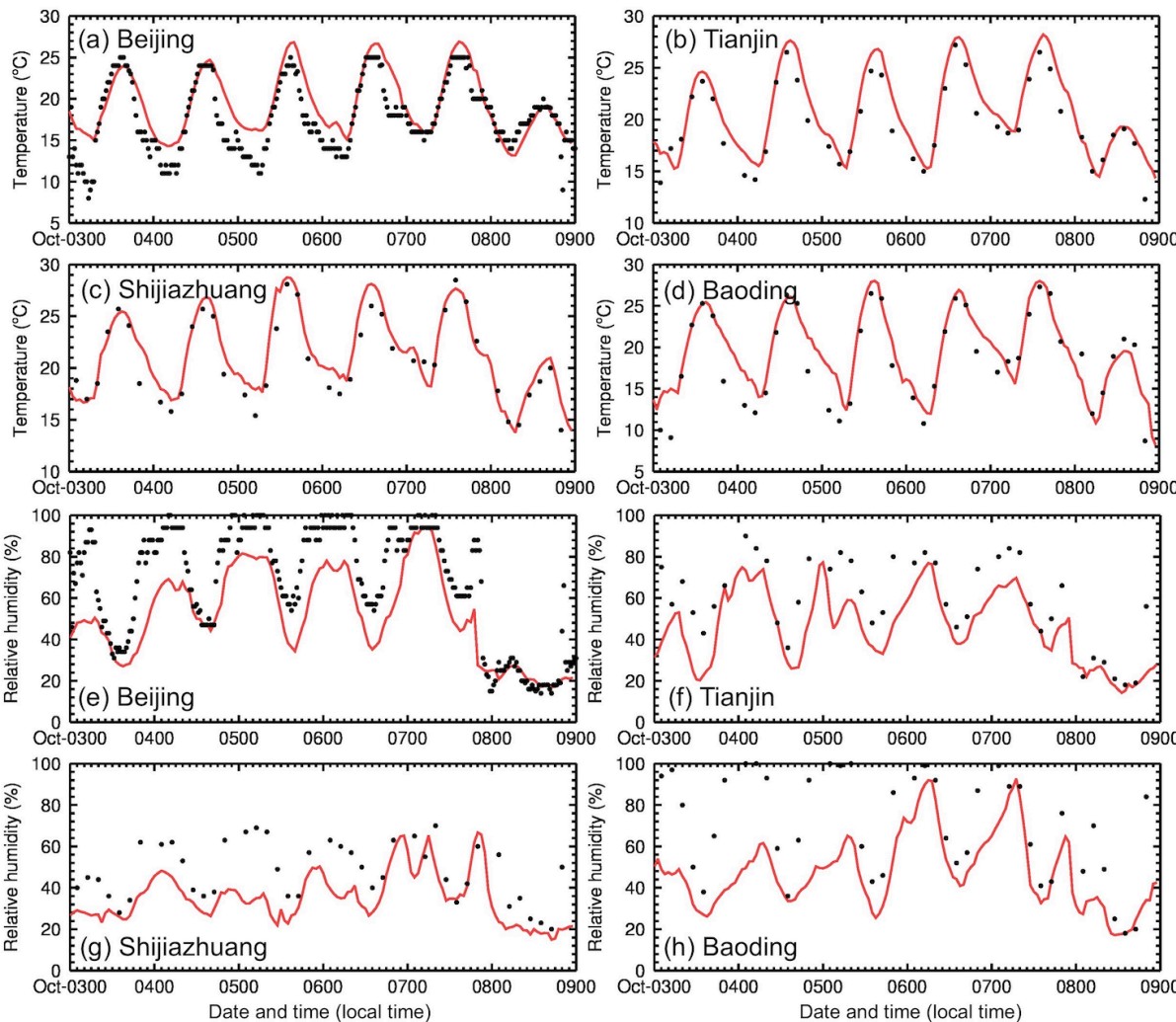



Figure 4 Simulated (red lines) and observed (black dots) temporal profiles of surface (a-d)
temperature and (e-h) relative humidity in (a, e) Beijing, (b, f) Tianjin, (c, g) Shijiazhuang,
and (d, h) Baoding from 3 to 8 October 2015.





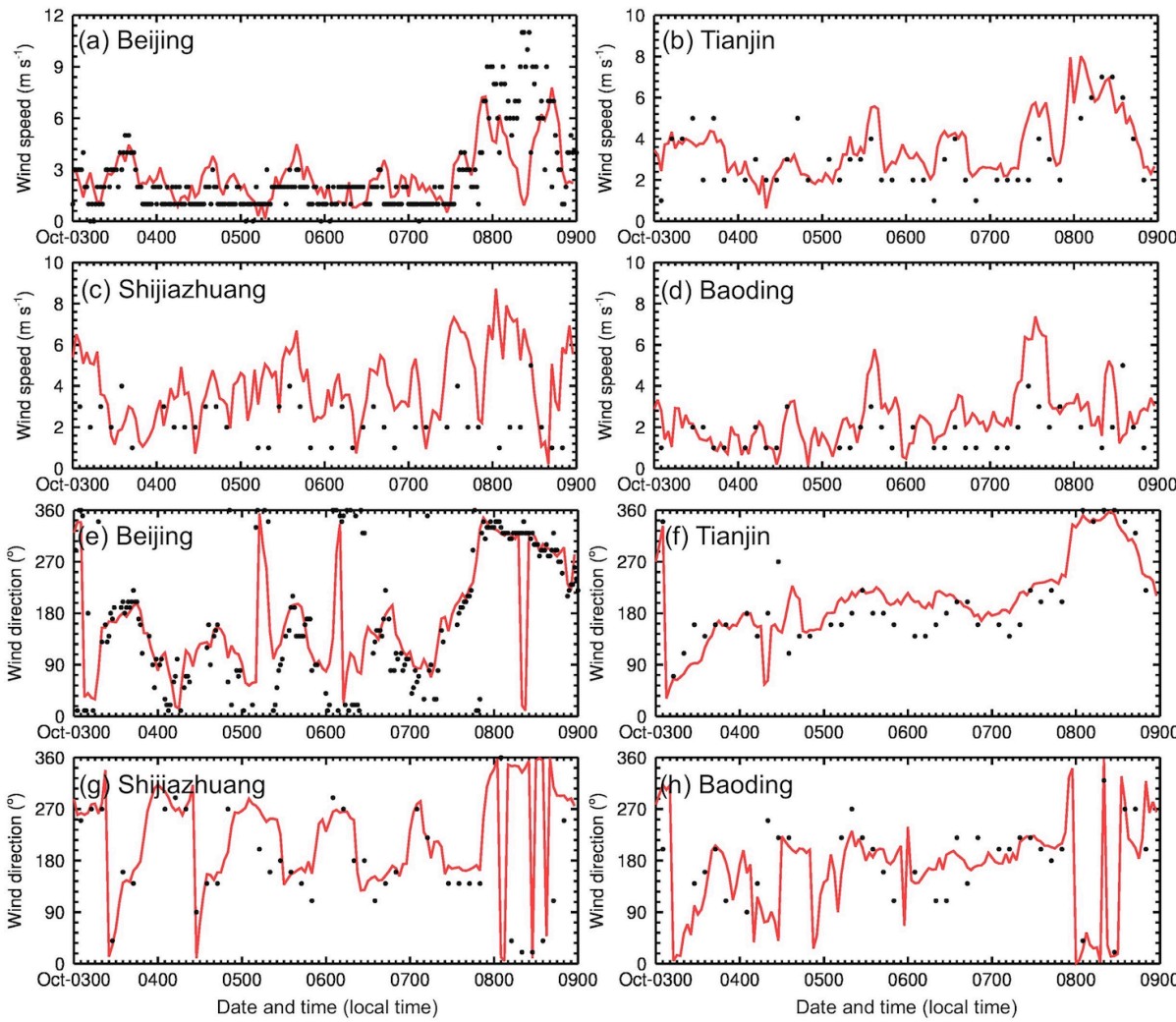



Figure 5 Simulated (red lines) and observed (black dots) temporal profiles of surface (a-d)
wind speed and (e-h) wind direction in (a, e) Beijing, (b, f) Tianjin, (c, g) Shijiazhuang, and
(d, h) Baoding from 3 to 8 October 2015.





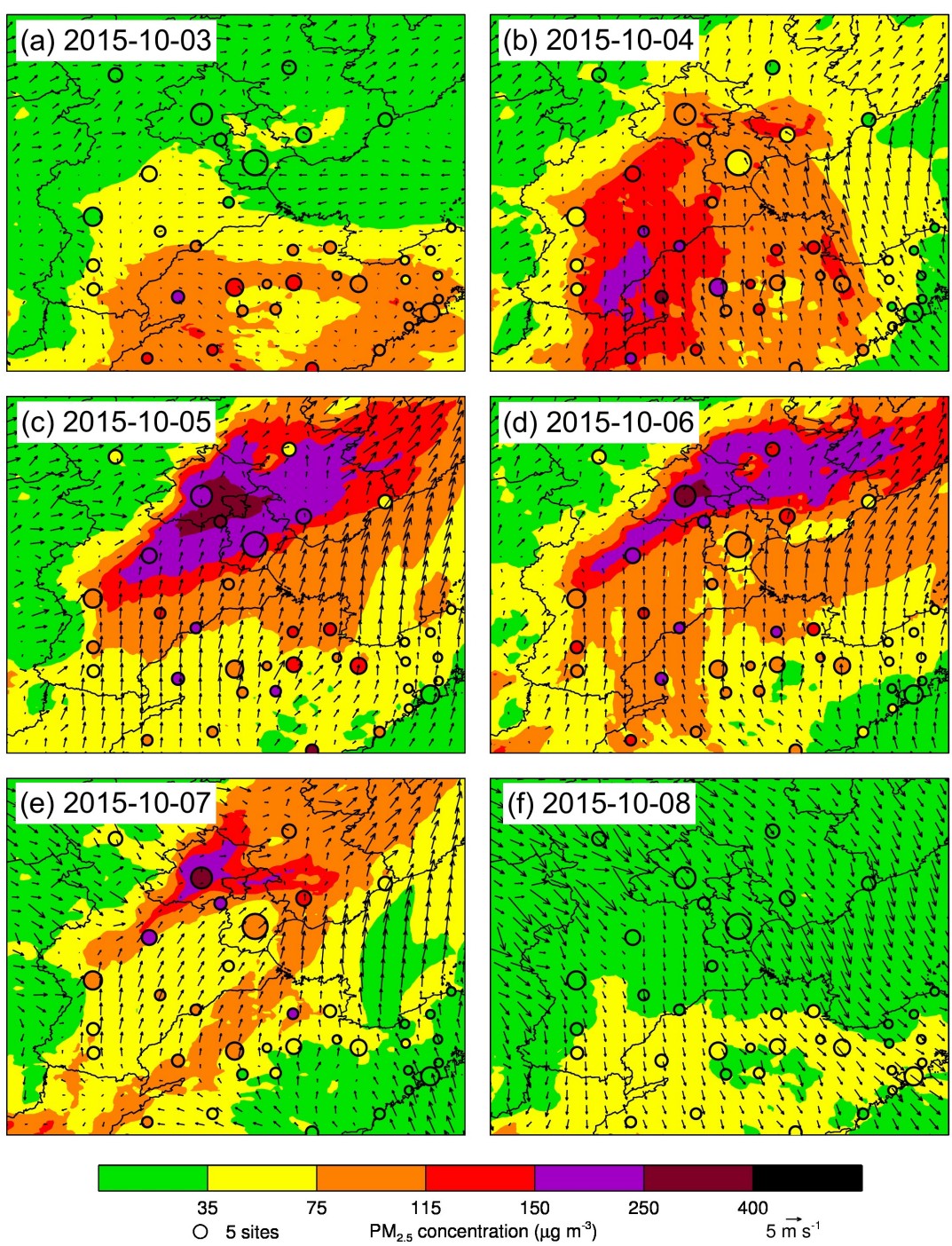

Figure 6 Spatial distributions of the modeled (colored shadings) and observed (colored dots) surface daily $PM_{2.5}$ concentration from 3 to 8 October 2015. Black arrows show the simulated surface winds.

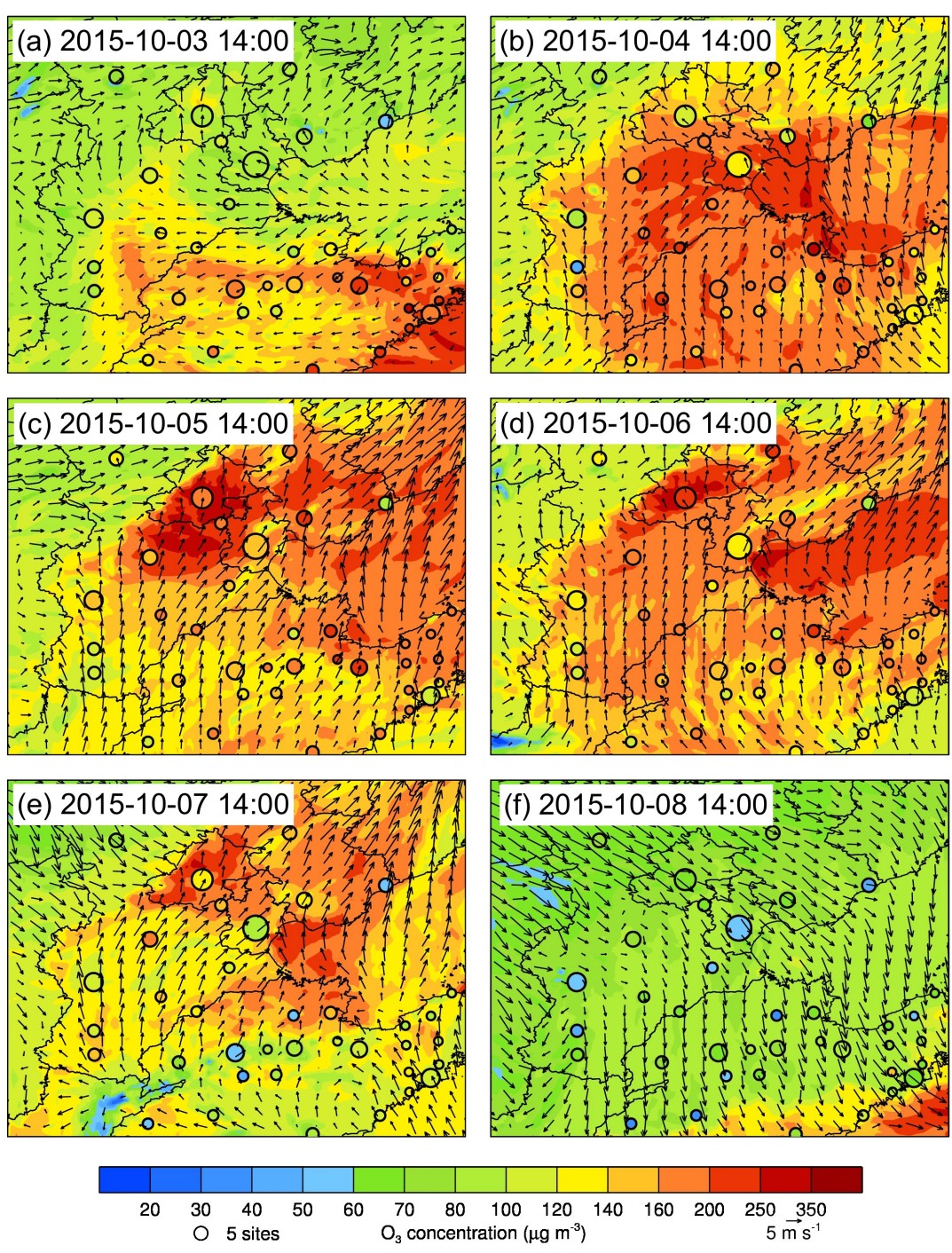

Figure 7 Spatial distributions of the modeled (colored shadings) and observed (colored dots) surface $O_3$ concentration at 14:00 (local time) from 3 to 8 October 2015. Black arrows show the simulated surface winds.

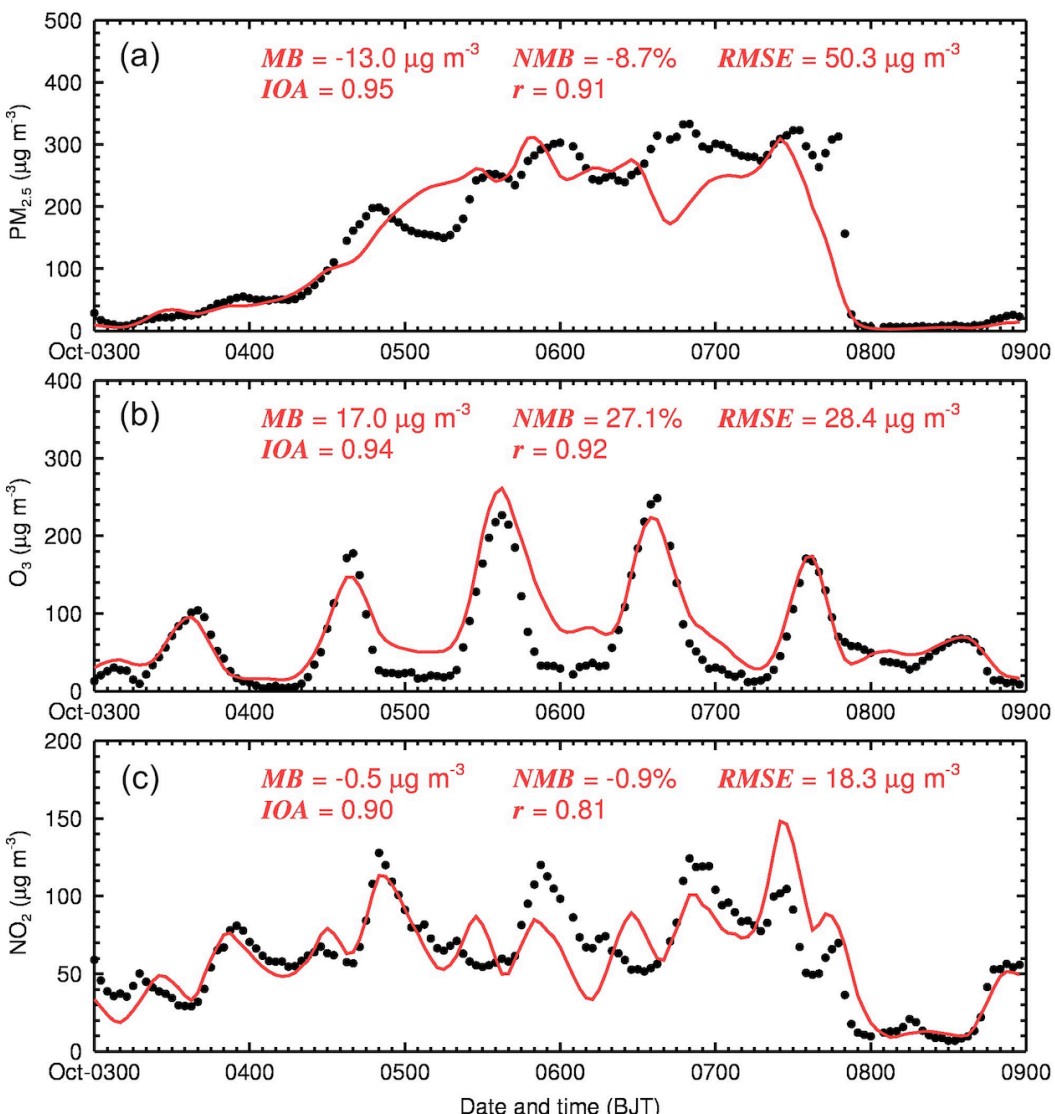

Figure 8 Diurnal variations of the modeled (red curves) and observed (black dots) surface (a)
PM$_{2.5}$, (b) O$_3$, and (c) NO$_2$ concentrations averaged over 12 ambient monitoring stations in
Beijing from 3 to 8 October 2015.

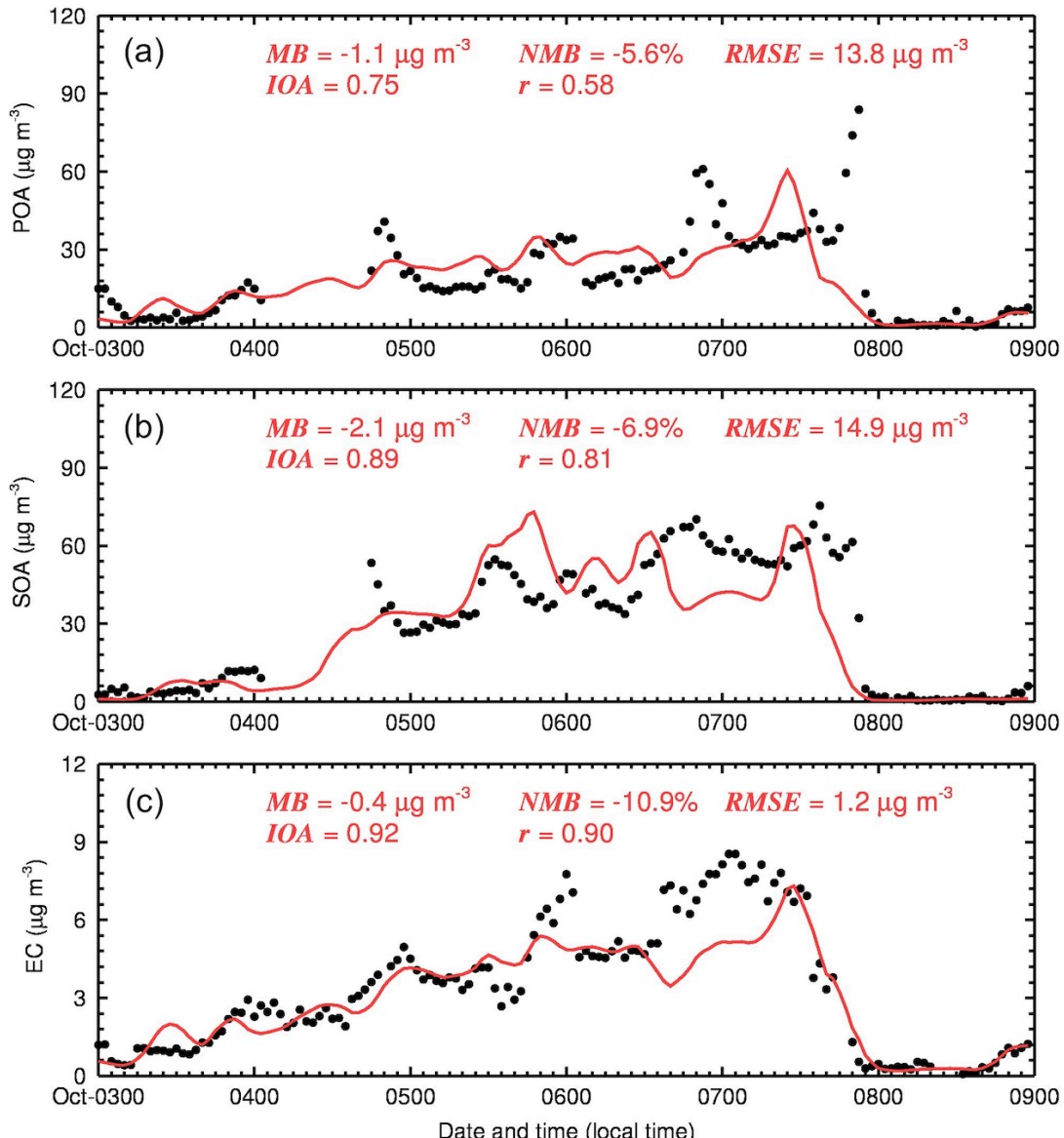

Figure 9 Diurnal variations of the modeled (red curves) and observed (black dots) surface
submicron (a) POA and (b) SOA concentrations at the NCNT station, and (c) EC
concentration in PM$_{2.5}$ at the CRAES station in Beijing from 3 to 8 October 2015.

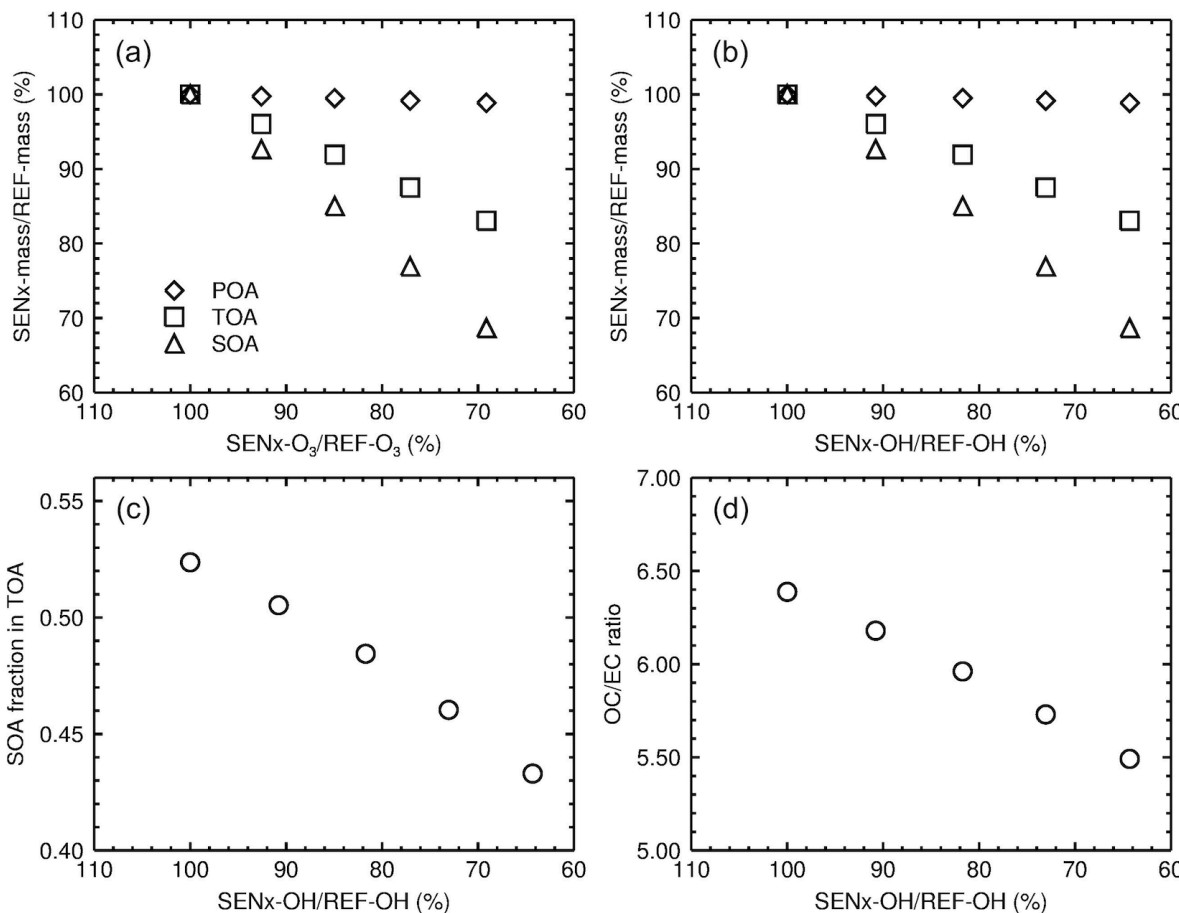



Figure 10 Impacts of changes in the AOC on organic aerosol components in BTH in 4
sensitivity experiments. (a) Concentration changes of POA, SOA, and TOA *versus* O₃, (b)
Concentration changes of POA, SOA, and TOA *versus* OH, (c) SOA fraction in TOA *versus*
OH concentration change, and (d) OC/EC ratio *versus* OH concentration change.





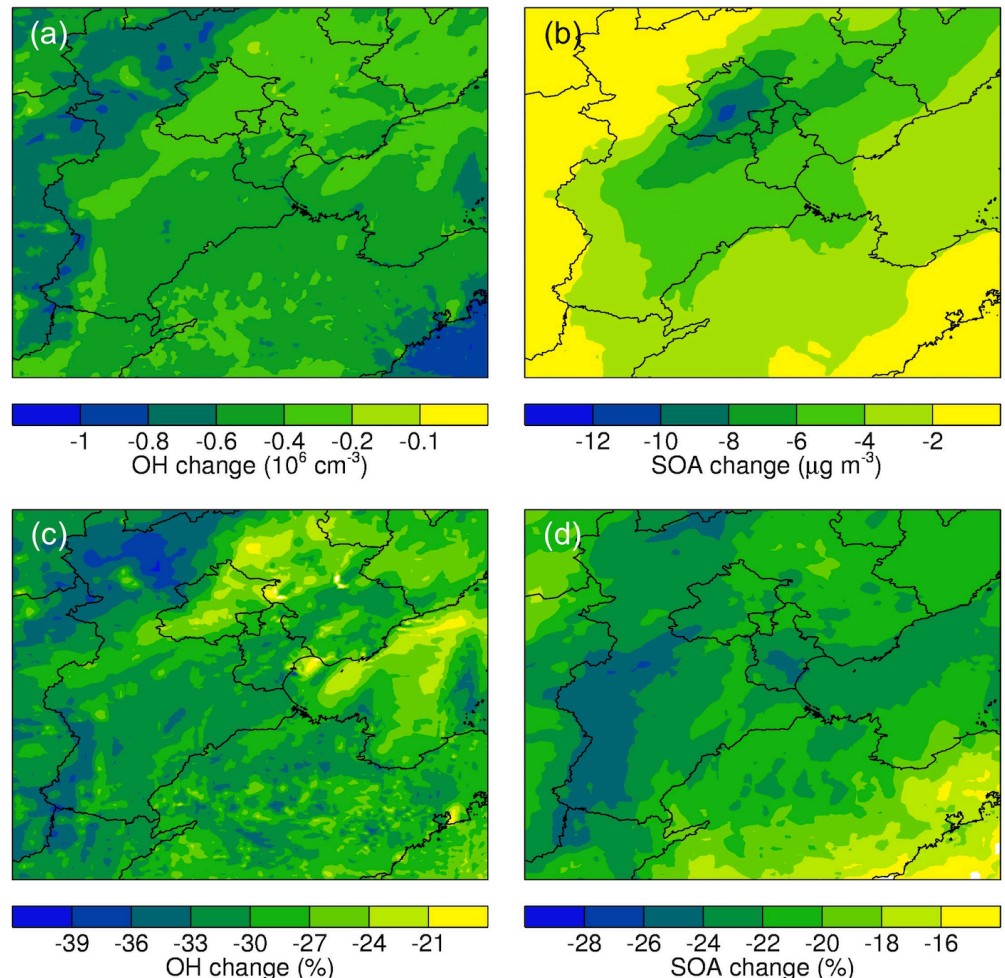



Figure 11 Spatial distributions of changes in (a, c) OH and (b, d) SOA concentrations
averaged from 4 to 7 October 2015 in the SEN3 experiment compared to the REF simulation
(SEN3 − REF).





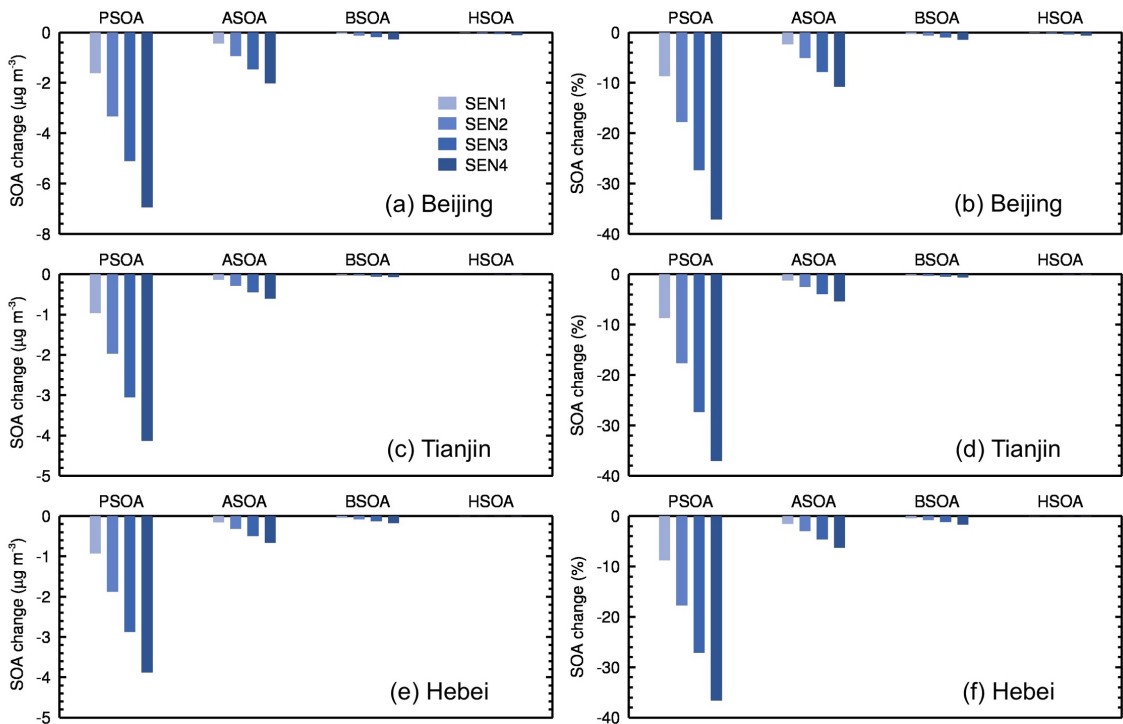

Figure 12 Histogram showing the decreases of SOA from various pathways in (a, b) Beijing,
(c, d) Tianjin, and (e, f) Hebei in the sensitivity experiments compared to the REF simulation
(SENx − REF, x = 1, 2, 3, and 4). PSOA: oxidation and partitioning of semivolatile POA and
co-emitted IVOCs; ASOA: oxidation and partitioning of anthropogenic VOCs; BSOA:
oxidation and partitioning of biogenic VOCs; HSOA: heterogeneous reactions of glyoxal and
methylglyoxal on aerosol surfaces.