# Peer review of "Secondary organic aerosol enhanced by increasing atmospheric oxidizing capacity in 1 2 Beijing-Tianjin-Hebei (BTH), China"

_Atmospheric Chemistry and Physics, 2018_

## Referee Comment (RC1) · Anonymous Referee #1 · 19 Feb 2019

The study provides a comprehensive analysis of the impact of chancing oxidative capacity of the atmosphere on SOA levels over the BTH region in China by using the WRF-Chem model over an episode. The study is very interesting, the experiments are well-designed and justified and the manuscript is well written and easy to follow, I enjoyed reading it. I have a number of comments and few technical/editorial corrections listed below before the manuscript can be published in ACP.

Comments:

Materials and Methods

Domain setup should be described. Is it only one domain that the simulations are

carried out or is this a inner domain of a nested domain system?

More details should be provided for the spatial, temporal and chemical distributions of anthropogenic emissions? Which profiles are used to if total PM and NMVOC emissions and annual emissions are used in the study? These can be already explained in another publication in detail but for the sake of having a stand-alone paper, authors could briefly explain these.

How about dust and biomass burning emissions?

I recommend to calculate also the correlation coefficient and Normalized Mean Bias (MNB) to provide relative changes in concentrations to be clearer the size of the changes to the readers outside China. Add these relative changes also in the text wherever you write about changes in mass.

It would also help to add a table summarizing the different scenarios. In addition, lines 247-254 fits better to the Materials and Methods sections.

Results

Lines 239-242: Can the underestimation be also due to the anthropogenic emissions?

Line 251: Change numbers from "reduced to" to "reduced by" to be consistent with the rest of the text (e.g. abstract, conclussions)

Lines 279-287: Discuss the reasons of the geographical differences (emissions, forests etc.).

Lines 301-304. Explain/discuss why largest impact is seen in PSOA pathway.

Technical corrections

Fig.1. Different to distinguish the different symbols.

Fig.2. Are these annual emissions?

Fig.12 caption, add the full names of PSOA, ASOA etc. in the caption.

Line 20: "...O3 concentrationS..."

Line 31: add units to 0.52 to 0.43.

Li et al., 2017a is not in the manuscript

Tie et al., 2016 is not in the manuscript

---

## Referee Comment (RC2) · Anonymous Referee #2 · 20 Feb 2019

The research presented in the paper aims to quantify the impacts of atmospheric oxidizing capacity (AOC) on the secondary organic aerosols (SOA) as well as ozone in North China. A specific version of WRF-Chem developed by the authors were employed to simulate the role of AOC during a winter severe haze event. I found the results from this work are critically important in understanding and evaluating the Air Pollution Control Action Plan currently being implemented in China. The paper is well organized and written, and the topic is highly relevant with the scope of ACP. Hence, I recommend acceptance of the manuscript after the following minor comments are addressed.

1) The simulated AOC influence on SOA is convincing, but one question unclear to me is what cause the AOC increase in observations? Is it due to the reduced aerosol concentration and elevated near-surface solar radiation after the pollution control plan?

2) The increase in the ratio of OC to EC has significant implication for atmospheric radiation and thermodynamical profiles. It may increase both aerosol scattering and absorption simultaneously. The former is related with near-surface solar radiation as mentioned above, while the latter further regulates the atmospheric stability. If the authors have related model output from the case study, it would interesting to examine those changes in the atmosphere.

3) L126-128, good to see a FDDA method is used in the model simulations. Can the authors be specific about what meteorological fields are constrained by what observations?

4) L169, why not from 2013 to 2017 like Fig. 3a? Is it due to the data availability?

5) Fig. 8&9, the model reasonably well reproduces the temporal evolutions of the major pollutants, both gaseous and particulate. The authors should mention what is the temporal resolution of the emission data used in this study and if it provides the data during the exact same time period.

6) L241-242, why do the wind biases cause an underestimation of EC only, not POA?

7) Figure 6, why the circles in the map have different sizes?

---

## Author Comment (AC1) · 29 Apr 2019

**Reply to Anonymous Referee #1**

We thank the reviewer for the careful reading of our manuscript and helpful comments. We have revised the manuscript following the suggestion, as described below.

The study provides a comprehensive analysis of the impact of changing oxidative capacity of the atmosphere on SOA levels over the BTH region in China by using the WRF-Chem model over an episode. The study is very interesting, the experiments are well-designed and justified and the manuscript is well written and easy to follow, I enjoyed reading it. I have a number of comments and few technical/editorial corrections listed below before the manuscript can be published in ACP.

**Materials and Methods**

**Comment 1.** Domain setup should be described. Is it only one domain that the simulations are carried out or is this an inner domain of a nested domain system?
**Response:** We have clarified in Section 2.1: "*The model is configured with one single domain which is centered at 116°E and 38°N with grid spacing of 6 km×6 km (200×200 grid cells). Thirty-five stretched vertical levels with spacing ranging from about 50 m near surface, to 500 m at 2.5 km, and 1 km above 14 km are used in the model configuration.*" in Lines 136-139.

**Comment 2.** More details should be provided for the spatial, temporal and chemical distributions of anthropogenic emissions. Which profiles are used to if total PM and NMVOC emissions and annual emissions are used in the study? These can be already explained in another publication in detail but for the sake of having a stand-alone paper, authors could briefly explain these.
**Response:** We have clarified in Section 2.1: "*The monthly average anthropogenic emission inventory is developed by Zhang et al. (2009) and Li et al. (2017c) with the base year of 2013, including agriculture, industry, power generation, residential, and transportation sources. The temporal resolution of emissions used in simulations is 1 hour, and the temporal allocation for different sources follows those in Zhang et al. (2009). Figure 2 presents the spatial distributions of anthropogenic volatile organic compounds (VOCs) and organic carbon (OC) emissions in October, showing high emissions in urban areas. The emissions of various species in Beijing,*

*Tianjin, Hebei, and the entire domain in October 2015 are summarized in Table 1.*" in Lines 139-147.

*Table 1 Anthropogenic emissions of various species in the simulation domain in October 2015 (Unit: Mton month$^{-1}$)*

| Species | NOx | SO$_2$ | NH$_3$ | CO | VOC | OC | EC |
|---------|------|------|------|-------|--------|------|------|
| Beijing | 0.31 | 0.02 | 0.05 | 0.66 | 1.51 | 0.03 | 0.01 |
| Tianjin | 0.24 | 0.09 | 0.05 | 0.09 | 2.8 | 0.05 | 0.01 |
| Hebei | 2.21 | 0.7 | 0.62 | 3.59 | 21.59 | 0.41 | 0.06 |
| Domain | 14.21 | 7.1 | 4.45 | 22.19 | 124.71 | 2.56 | 0.3 |

**Comment 3.** How about dust and biomass burning emissions?

**Response:** We have clarified in Section 2.1: "*The GOCART (Georgia Tech/Goddard Global Ozone Chemistry Aerosol Radiation and Transport model) dust module is used to estimate the emission, transport, dry deposition, and gravitational settling of dust (Ginoux et al., 2001). The biomass burning emissions are from the Fire Inventory from NCAR (FINN) (Wiedinmyer et al., 2011; 2006).*" in Lines 125-129, and the references are updated.

*Ginoux, P., Chin, M., Tegen, I., Prospero, J. M., Holben, B., Dubovik, O. and Lin, S.-J.: Sources and distributions of dust aerosols simulated with the GOCART model, J. Geophys. Res., 106(D17), 20255–20273, doi:10.1029/2000JD000053, 2001.*

*Wiedinmyer, C., Akagi, S. K., Yokelson, R. J., Emmons, L. K., Al-Saadi, J. A., Orlando, J. J., and Soja, A. J.: The Fire INventory from NCAR (FINN): a high resolution global model to estimate the emissions from open burning, Geosci. Model. Dev., 4, 625-641, doi: 10.5194/gmd-4-625-2011, 2011.*

*Wiedinmyer, C., Quayle, B., Geron, C., Belote, A., McKenzie, D., Zhang, X., O'Neill, S., and Wynne, K. K.: Estimating emissions from fires in North America for air quality modeling, Atmos. Environ., 40, 3419-3432, doi: 10.1016/j.atmosenv.2006.02.010, 2006.*

**Comment 4.** I recommend to calculate also the correlation coefficient and Normalized Mean Bias (NMB) to provide relative changes in concentrations to be clearer the size of the changes to the readers outside China. Add these relative changes also in the text wherever you write about changes in mass.

**Response:** We have included the correlation coefficient (*r*) and normalized mean bias (NMB) and implemented them in discussions. Figures 8 and 9 have been updated accordingly.

In Section 2.4 (Lines 168-179):

"    The mean bias (MB), normalized mean bias (NMB), root mean square error (RMSE), index of agreement (IOA), and linear Pearson correlation coefficient (r) are selected to evaluate the WRF-Chem model simulations against observations.

$$MB = \frac{1}{N}\sum_{i=1}^{N}(P_i - O_i)\qquad(1)$$

$$NMB = \frac{\sum_{i=1}^{N}(P_i-O_i)}{\sum_{i=1}^{N}O_i} \times 100\%\qquad(2)$$

$$RMSE = \left[\frac{1}{N}\sum_{i=1}^{N}(P_i - O_i)^2\right]^{\frac{1}{2}}\qquad(3)$$

$$IOA = 1 - \frac{\sum_{i=1}^{N}(P_i-O_i)^2}{\sum_{i=1}^{N}(|P_i-\bar{O}|+|O_i-\bar{O}|)^2}\qquad(4)$$

$$r = \frac{\sum_{i=1}^{N}(P_i-\bar{P})(O_i-\bar{O})}{\sqrt{\sum_{i=1}^{N}(P_i-\bar{P})^2}\sqrt{\sum_{i=1}^{N}(O_i-\bar{O})^2}}\qquad(5)$$

where $P_i$ and $O_i$ are the simulated and observed variables, respectively. $N$ is the total number of predictions. $\bar{P}$ and $\bar{O}$ denote the average of predictions and observations, respectively. IOA ranges from 0 to 1 theoretically, with 1 suggesting perfect agreement between predictions and observations."

In Section 3.2.2 (Lines 249-257):

"The model generally replicates the evolution of the observed $PM_{2.5}$ concentration with an IOA (r) of 0.95 (0.91), but slightly underestimates the $PM_{2.5}$ concentration with an MB (NMB) of -13.0 $\mu g\ m^{-3}$ (-8.7%). The simulated diurnal profile of the $O_3$ concentration is well consistent with observations, with an IOA (r) of 0.94 (0.92), but the model overestimates the $O_3$ diurnal lows during the maturation stage. Additionally, Figures 8a and 8b also show that both $O_3$ and $PM_{2.5}$ pollutions occur during the maturation stage in Beijing, as previously reported for non-winter seasons (Jia et al., 2017). The model also exhibits good performance in simulating the temporal variation of $NO_2$ concentrations, with an IOA (r) of 0.90 (0.81)."

In Section 3.2.3 (Lines 262-270):

"The model yields the increasing trend of the POA concentration from the startup to maturation stages compared to the measurements, but cannot well capture the observed spiky peaks, with an IOA (r) of 0.75 (0.58). Figure 9b shows that the observed SOA concentration is remarkably enhanced during the maturation stage, ranging from 30 to 90 $\mu g\ m^{-3}$, which is well predicted by the model. The MB, NMB, IOA, and r for the simulated SOA concentration are -2.1 $\mu g\ m^{-3}$, -6.9%, 0.89, and 0.81, respectively. Although the IOA and r for the simulated EC concentration

*reach 0.92 and 0.90, respectively, the model considerably underestimates the EC concentration against measurement on October 6 and 7, which is likely caused by the variation in the anthropogenic emissions."*

**Comment 5.** It would also help to add a table summarizing the different scenarios. In addition, lines 247-254 fits better to the Materials and Methods sections.

**Response:** We have included a table summarizing the different scenarios in Table 3 and also moved the descriptions of the simulations to Section 2.3 (Lines 160-166):

*"2.3 Model simulations*

*We define the simulation with the AOC in October 2015 as the reference (REF). The model result in REF is compared with the observations to evaluate the model performance. To examine the impact of increasing AOC on OA components, we perform 4 sensitivity experiments (SEN1~4) by varying AOC. Compared with the REF simulation, we decrease all the photolysis frequencies by 10%, 20%, 30%, and 40%, respectively, in the model simulations."*

*Table 3 Description of the reference simulation and sensitivity experiments*

| Case ID | Description |
| --- | --- |
| REF | The reference simulation constrained by observations |
| SEN1 | 10% decrease in photolysis frequencies |
| SEN2 | 20% decrease in photolysis frequencies |
| SEN3 | 30% decrease in photolysis frequencies |
| SEN4 | 40% decrease in photolysis frequencies |

**Results**

**Comment 6.** Lines 239-242: Can the underestimation be also due to the anthropogenic emissions?

**Response:** We have revised the sentence as: "*Although the IOA and r for the simulated EC concentration reach 0.92 and 0.90, respectively, the model considerably underestimates the EC concentration against measurements on October 6 and 7, which is likely caused by the variation in anthropogenic emissions.*" in Lines 267-270.

**Comment 7.** Line 251: Change numbers from "reduced to" to "reduced by" to be consistent with the rest of the text (e.g. abstract, conclusions)

**Response:** We have changed the numbers from "*reduced to*" to "*reduced by*" to make it consistent throughout the text in Section 3.3: "*Compared to the REF simulation, when the photolysis frequencies are decreased by 10%, 20%, 30%, and 40% in the 4 sensitivity experiments (SEN1~4), respectively, the $O_3$ (OH radical) concentration is correspondingly reduced by 7.4% (9.2%), 15.1% (18.3%), 22.9% (26.9%), and 30.9% (35.7%). It is worth noting that the REF experiment is assumed to represent a situation in autumn with the high AOC, and the SEN1~4 experiments could be regarded as 4 scenarios with the different lower AOC.*" in Lines 275-280.

**Comment 8.** Lines 279-287: Discuss the reasons of the geographical differences (emissions, forests etc.)

**Response:** We have clarified in Section 3.3.3: "*Although OH is the main oxidant in the SOA formation during daytime, the spatial change of SOA concentration is not well consistent with that of the OH concentration, especially for the mass change (Figure 11a). The geographical difference probably results from the spatial distribution variation of anthropogenic and biogenic precursors of SOA. In the middle and east BTH, massive anthropogenic SOA precursors are emitted from residential, transportation and industrial sources; while in the west BTH, biogenic precursor emissions are dominant for the SOA formation, but much less than those from anthropogenic sources in the middle and east BTH (Figure 2).*" in Lines 318-326.

**Comment 9.** Lines 301-304. Explain/discuss why the largest impact is seen in PSOA pathway.

**Response:** We have clarified in Section 3.3.3: "*Since the oxidation and partitioning of semivolatile POA and co-emitted IVOCs contribute the most to the SOA concentration (Feng et al., 2016), the most substantial SOA decrease occurs in the PSOA, followed by the ASOA and BSOA.*" in Lines 340-342.

**Technical corrections**

**Comment 10.** Fig.1. Different to distinguish the different symbols.

**Response:** We have revised Figure 1 to make different symbols clearer.

[Figure]

*Figure 1 Model domain with the topography. The black circles denote the locations of the cities with ambient air quality monitoring sites, and the size of the circles represents the number of sites in each city. The white triangles show the location of the meteorological stations in Beijing, Tianjin, Shijiazhuang, and Baoding. The light blue and pink dots in Beijing show the observation sites with the POA/SOA (NCNT) and OC/EC (CRAES) measurements, respectively.*

**Comment 11.** Fig.2. Are these annual emissions?

**Response:** Fig.2 shows the emission distribution in October and we have revised the caption of Figure 2: "*Figure 2 Geographic distributions of monthly average anthropogenic emissions of (a) VOCs and (b) organic carbon in October in the simulation domain. The black lines present provincial boundaries in China.*" in Lines 745-747 and 686-688.

**Comment 12.** Fig.12 caption, add the full names of PSOA, ASOA etc. in the caption.

**Response:** We have revised the caption of Figure 12: "*PSOA: oxidation and partitioning of semivolatile POA and co-emitted IVOCs; ASOA: oxidation and partitioning of anthropogenic VOCs; BSOA: oxidation and partitioning of biogenic VOCs; HSOA: heterogeneous reactions of glyoxal and methylglyoxal on aerosol surfaces.*" in Lines 845-848 and 721-724.

**Comment 13.** Line 20: ". . .O$_3$ concentrations. . ."

**Response:** We have changed "*O$_3$ concentration*" to "*O$_3$ concentrations*" in Line 22.

**Comment 14.** Line 31: add units to 0.52 to 0.43.

**Response:** We have added "*dimensionless*" in Lines 33-34: "*... the SOA fraction in total organic aerosol by 17% (from 0.52 to 0.43, dimensionless) ...*" to make it clearer.

**Comment 15.** Li et al., 2017a is not in the manuscript

**Response:** The citation of Li et al., 2017a is in Lines 45-46.

**Comment 16.** Tie et al., 2016 is not in the manuscript

**Response:** The citation of Tie et al., 2016 is in Lines 48-49.

---

## Author Comment (AC2) · 29 Apr 2019

**Reply to Anonymous Referee #2**

We thank the reviewer for the careful reading of our manuscript and helpful comments. We have revised the manuscript following the suggestion, as described below.

The research presented in the paper aims to quantify the impacts of atmospheric oxidizing capacity (AOC) on the secondary organic aerosols (SOA) as well as ozone in North China. A specific version of WRF-Chem developed by the authors were employed to simulate the role of AOC during a winter severe haze event. I found the results from this work are critically important in understanding and evaluating the Air Pollution Control Action Plan currently being implemented in China. The paper is well organized and written, and the topic is highly relevant with the scope of ACP. Hence, I recommend acceptance of the manuscript after the following minor comments are addressed.

**Comment 1.** The simulated AOC influence on SOA is convincing, but one question unclear to me is what cause the AOC increase in observations? Is it due to the reduced aerosol concentration and elevated near-surface solar radiation after the pollution control plan?

**Response:** We have clarified in Section 3.1: "*The reason for the AOC or O₃ increase since 2013 still remains elusive. Li et al. (2018) have proposed that the O₃ increase in China since 2013 is associated with the decreased removal efficiency of $HO_x$ (OH + peroxy) on aerosol surfaces caused by the reduced aerosol concentrations since the implementation of APPCAP. However, further studies need to be conducted to evaluate the O₃ contribution of the photolysis change caused by the aerosol-radiation interaction and aerosol-cloud interaction induced by decreasing aerosols in China.*" in Lines 191-197.

*Li, K., Jacob, D. J., Liao, H., Shen, L., Zhang, Q. and Bates, K. H.: Anthropogenic drivers of 2013-2017 trends in summer surface ozone in China, Proc. Natl. Acad. Sci. U.S.A., 17, 201812168–6, doi:10.1073/pnas.1812168116, 2018.*

**Comment 2.** The increase in the ratio of OC to EC has significant implication for atmospheric radiation and thermodynamical profiles. It may increase both aerosol scattering and absorption simultaneously. The former is related with near-surface solar radiation as mentioned above, while the latter further regulates the atmospheric stability. If the authors have related model output from the case study, it would be interesting to examine those changes in the atmosphere.

**Response:** We have clarified in Section 3.3.1: "*It is worth noting that the increase in OC/EC ratio potentially influences atmospheric radiation and thermodynamical profiles, through enhancing aerosol scattering and absorption simultaneously (Wang et al., 2013). When the photolysis frequencies are reduced by 30% in the SEN3 experiment, compared to the REF, the downward shortwave radiation is reduced by 1.2 W m$^{-2}$ on average in BTH, and the surface temperature is decreased by around 0.016 $^o$C during daytime. Effects of the AOC change on the temperature profile is not significant, and the daytime temperature decrease in the SEN3 experiment is less than 0.005 $^o$C within 1 km height from surface.*" in Lines 297-304.

**Comment 3.** L126-128, good to see a FDDA method is used in the model simulations. Can the authors be specific about what meteorological fields are constrained by what observations?

**Response:** We have revised the sentence to make it clearer in the text: "*Specifically, the surface and upper air observational wind fields from China Meteorological Administration (CMA) during the study period are assimilated using the four-dimensional data assimilation (FDDA) method to better simulate meteorological fields.*" in Lines 131-134.

**Comment 4.** L169, why not from 2013 to 2017 like Fig. 3a? Is it due to the data availability?

**Response:** Yes, the data after 2015 are not available.

**Comment 5.** Fig. 8&9, the model reasonably well reproduces the temporal evolutions of the major pollutants, both gaseous and particulate. The authors should mention what is the temporal resolution of the emission data used in this study and if it provides the data during the exact same time period.

**Response:** We have clarified in Section 2.1: "*The monthly average anthropogenic emission inventory is developed by Zhang et al. (2009) and Li et al. (2017c) with the base year of 2013, including agriculture, industry, power generation, residential, and transportation sources. The temporal resolution of emissions used in simulations is 1 hour, and the temporal allocation for different sources follows those in Zhang et al. (2009).*" in Lines 139-144.

**Comment 6.** L241-242, why do the wind biases cause an underestimation of EC only, not POA?

**Response:** We have revised the sentence as: "*Although the IOA and r for the simulated EC concentration reach 0.92 and 0.90, respectively, the model considerably underestimates the EC concentration against measurements on October 6 and 7, which is likely caused by the variation in anthropogenic emissions.*" in Lines 267-270.

**Comment 7.** Figure 6, why the circles in the map have different sizes?

**Response:** We have clarified in the caption of Figure 1: "*... the size of the circles represents the number of sites in each city.*" in Lines 733-734 and 681-682.